# Efficacy of Aromatherapy at Relieving the Work-Related Stress of Nursing Staff from Various Hospital Departments during COVID-19

**DOI:** 10.3390/healthcare11020157

**Published:** 2023-01-04

**Authors:** Chi-Lun Hung, Yun-Ling Lin, Chin-Mei Chou, Ching-Ju Wang

**Affiliations:** 1Department of Industrial Engineering and Management, Yuan Ze University, Taoyuan 32003, Taiwan; 2Taipei Veterans General Hospital, Hsinchu Branch, Taipei 112, Taiwan

**Keywords:** aromatherapy, bergamot essential oil, nursing-related stress

## Abstract

This study aimed to evaluate the efficacy of aromatherapy in relieving the stress of nursing staff working in different departments during COVID-19. A total of 26 nursing staff from Taiwan were recruited for this study. Bergamot essential oil was diffused for over a four-week period in four different hospital departments. We assessed heart rate variability indicators, Nurse Stress Checklist, and Copenhagen Burnout Inventory before and after the intervention. The results of the analysis showed that during a high workload period, aromatherapy had no significant effect on regulating physical stress. Subjective measurements showed a significant impact on work concern and personal fatigue. Moreover, there were large differences among the four departments; the aromatherapy treatment had a weak effect on those with a heavy workload, whereas those with a lighter workload showed a stronger effect. Finally, this study provides practical results about aromatherapy stress reduction applied during the pandemic on first-line medical staff.

## 1. Introduction

The nature of the work of nursing staff includes a 24-h shift system and the responsibility of dealing with demanding job requirements, medical disputes, emotional stress, adaptation to staff transfer, and other work-related stress, which make them a high-risk group for workplace fatigue [1,2,3]. Previous studies have concluded that high levels of work-related stress affect the job performance of nursing staff and the quality of care they provide, resulting in more incidents of medical negligence. Long-term stress also causes other negative effects such as nervousness, anxiety, and insomnia, thereby reducing the quality of life of affected individuals [4,5].

According to Jang et al. [6], the level of physical and mental fatigue experienced by nursing staff is significantly affected by the hospital departments in which they work. Those working in specialised departments such as the emergency room (ER), intensive care unit (ICU), operating room (OR), and post-anaesthesia care unit (PACU) experience higher levels of physical and mental fatigue than their colleagues who work in general departments such as internal medicine, general surgery, and obstetrics and gynaecology (ObGyn). Warshawsky and Havens [7] surveyed 291 nursing staff working in the ER and found that approximately 62% planned to resign in 2–5 years because of the high amounts of work-related stress. Additionally, high-intensity work-related stress was found to cause high turnover among nursing staff. In a survey of surgical nurses by Bulbuloglu et al. [8], 45.5% of the respondents reported being stressed, and as many as 80% believed they were negatively affected by their work environment. Thus, it is important to find ways to effectively relieve the stress faced by nursing staff in the workplace.

Medical and nursing staff began experiencing severe anxiety, stress, and post traumatic stress disorder symptoms with the global impacts of COVID-19 [9]. In addition to their usual responsibilities in providing medical care, the nurses’ workload increased substantially because of limited medical resources, which worsened their work-related stress [10,11]. González-Gil et al. [12] conducted a questionnaire survey among the ICU and ER nursing staff during the peak of the COVID-19 pandemic and found that they were prone to psychological and emotional issues in the short and medium terms, because of their concerns over viral infections. Another study showed that the psychological stress faced by healthcare workers working in non-COVID departments was significantly higher than that of COVID-19 frontline healthcare workers [13]. This indicated that all nursing staff had to bear additional physical and psychological stress and worse fatigue levels during the pandemic compared with their pre-pandemic routines, regardless of whether they had direct contact with COVID-19 patients.

Currently, common symptoms of stress can be managed through either drug therapy or psychotherapy. The latter includes music therapy [14,15], relaxation therapy [16,17], light or phototherapy [18], and aromatherapy [19]. Among these, aromatherapy is a form of complementary therapy. It operates as follows: the essential oil molecules released during aromatherapy are inhaled through the nose, enter the brain, and stimulate and activate the parasympathetic nervous system [20], thereby achieving the effects of sedation and stress relief [21]. Aromatherapy is a convenient and uncomplicated method of stress intervention, and the treatment can be done through various methods, including direct inhalation, massage, and diffusion. However, the duration of the treatment effects depends on the type of intervention [22]. Chen et al. [23] indicated that when diffusion is used, aromatherapy as an intervention must be sustained for 3–4 days before it starts producing effects. However, direct inhalation shows effects in only 10–15 min [22].

Considering the constraints of the work environments, aromatherapy is often used to relieve the stress and anxiety of nurses. Lavender essential oil has been verified to produce stress-relieving effects in many medical spaces such as the ER and ICU [24,25] and nursing stations [23]. Bergamot essential oil was also found to produce similar effects: after intervention with the essential oil, work-related stress was relieved [26,27] and sleep problems were alleviated [28]. In a past study, Watanabe et al. [29] observed that the experimental group experienced simultaneous improvements in their physical state and subjective mood following the use of bergamot as an intervention. Bergamot essential oil contains Linalool (6–15%), an analgesic [30,31] that has anti-anxiety, anti-depression, and neuroprotective properties [32], and Linalyl acetate (23–35%) [33,34,35]. Because of these effects, it has also been frequently applied to clinical patients to delay pain [36] and reduce anxiety [37]. Additionally, citrus essential oils—such as grapefruit, sweet orange, and bergamot—contain limonene (30–50%), which has pain relief and anti-inflammation effects [38]. In summary, bergamot essential oil can stimulate the parasympathetic nervous system and has mild antibacterial effects, making it suitable for application in environments where viruses are rampant during the pandemic. Moreover, its aroma is very similar to that of other citruses, so it was safe to use [39].

Although several studies have highlighted the efficacy of aromatherapy in relieving the stress of nurses, few have explored pandemic-related nursing stress. This study was conducted during the second COVID-19 outbreak in Taiwan (April–June 2021). During this period, diffusers were placed in different work departments of a particular hospital to provide aromatherapy to the nursing staff and observe their physical status before and after the intervention. Previous studies on aromatherapy mostly used subjective questionnaires to evaluate its effects before and after the intervention. However, other studies have also suggested that the pre- and post-intervention responses of the parasympathetic nervous system are more accurately evaluated based on physiological values such as heart rate (HR) and heart rate variability (HRV) [22,23]. Therefore, the HRV indicators and a subjective questionnaire were both used in this study to evaluate the efficacy of aromatherapy to relieve the stress of nursing staff working in different departments.

## 2. Materials and Methods

### 2.1. Participants

The participants of this study were 30 existing nursing staff members of the Taipei Veterans General Hospital (Hsinchu Branch) who worked in the hospital’s general, palliative care, and ObGyn departments, as well as the ICU. The inclusion criteria for participation were: (i) had a minimum of three months’ experience in clinical practice and possessed a licence to practice; (ii) did not suffer from any special diseases (heart disease or asthma or other respiratory diseases); and (iii) maintained their normal work schedule over the entire 30-day experimental period (participants who subsequently resigned or had to be isolated were excluded from the experiment).

The study was reviewed and approved by the Institutional Review Board (IRB) of the Taipei Veterans General Hospital (IRB No. 2021-03-005CC). The experimental procedures were explained to the participants prior to the commencement of the study. They were also informed that they had the right to withdraw from the study at any time. They completed a consent form to participate in the study.

### 2.2. Experimental Procedures

Cho and Kwon indicated that aromatherapy by ambient delivery needs a longer time than inhaling to get a positive impact (for a total of 12–24 h) [22]. In our study, we used a diffuser and conducted aromatherapy consecutively for one month (four work weeks), with interventions made twice every weekday (Monday to Friday) at 08:00–12:00 a.m. and 16:00–20:00 p.m. The scent was 100% pure bergamot peel essential oil produced by AndZen Co., Sydney, Australia, and mainly contained limonene (<46%), linalyl acetate (<35%), and linalool (<23%). An ultrasonic diffuser was used for aroma evaporation (Rose Crown Co., Queensland, Australia), which only diffuses the pure essential molecules without water dilution. For each treatment session, five drops of pure bergamot oil were added, which made the diffuser constantly work for four hours, and the aroma molecules were dispersed into the environment through vibration. The interventions were conducted at the nursing stations of three types of spaces: general and palliative care departments (Figure 1a), ObGyn department (Figure 1b), and ICUs (Figure 1c). The nature of the nursing staff’s work required them to move around the entire space; however, most of their main work was done in the nursing station. Thus, the diffusers were placed at the nursing stations to ensure that all the personnel could experience the intervention. In addition, all the diffusers were placed at the back side of the nursing counter to prevent undesirable effects on patients.

### 2.3. Experimental Procedures

A quasi-experiment was conducted to investigate the effect of bergamot oil on nursing staff. The efficacy of aromatherapy as the experimental intervention was evaluated by collecting the participants’ physiological values and responses to a subjective questionnaire. The overall experimental flow chart is shown in Figure 2. First, pre-experimental records (pre) were made for all the participants one day prior to the intervention. After the completion of four weeks of intervention, the participants’ physiological data were recorded one day after the intervention ended. Participants were also asked to complete a questionnaire. The data were then used for post-experimental evaluation (post).

### 2.4. Physiological Parameters

Dzedzickis et al. [40] defined the physiological signals intended for the recognition of emotions as satisfying the condition of ‘non-conscious control’. HR is affected by the autonomic nervous system: it increases when people feel stressed or experience positive emotions but decreases when they are relaxed or bear negative emotions [41]. HRV-related signals are also often used as indicators to evaluate activities of the autonomic nervous system [42], and characteristics of the frequency domain of the signals can be extracted using Fourier transform. For example, 0.04–0.15 and 0.15–0.4 Hz are parameters indicating HR with low frequency powers (LF) and high frequency powers (HF), respectively. The regulatory responses of the sympathetic and parasympathetic nervous systems are distinguished by calculating the LF/HF ratio [43]. The physiological values, which included two blood pressure parameters and seven HRV parameters, were measured using the TS-0411 HRV analyser. Table 1 presents the clinical significance of each indicator.

### 2.5. Subjective Measures

The questionnaire comprised three major parts. The first part covered sociodemographic data including age, education level, work experience, nursing grade, and which hospital department they worked in. The other two parts included the Nurse Stress Checklist (NSC) and Copenhagen Burnout Inventory (CBI), which were used as the bases for subjective evaluations of nursing stress.

The NSC was used to evaluate the work-related stress of the nursing staff. The higher the total score, the greater the work-related stress. This tool could effectively assess the physical, psychological, and cognitive effects of stress in the medical environment; conflicts in the role of individuals; concerns over goals for personal care; job performance satisfaction; and personal work abilities [44]. The NSC (Chinese version) has been verified to have an overall Cronbach’s alpha that ranges from 0.93 [45] to 0.95 [46], indicating that the translated version was sufficiently effective for assessing nurses’ work-stress. There were 43 questions covering four aspects, as follows. Personal response (16 items), Work concerns (13 items), Competence (11 items), and Incompleteness of personal arrangement (3 items) were scored using a 7-point Likert scale. The Cronbach’s alpha for the four subscales in our assessment were 0.945, 0.911, 0.918, and 0.865, respectively.

The CBI was used to evaluate the feeling of burnout at work and when helping others, and its Cronbach’s alpha coefficient was 0.85–0.87. It included 19 questions and covered three dimensions for evaluation, namely, personal burnout, work-related burnout, and client-related burnout. A 5-point Likert scale was used for scoring. The higher the total score, the greater the degree of fatigue [47]. With a Cronbach’s alpha of 0.90, the reliability of the CBI (Chinese version) was verified as being equally effective after translation [48,49]. In this study, we only investigated questions about the subscales of personal burnout and client-related burnout, and the sum score of two questions. The Cronbach’s alpha for these subscales were 0.90 and 0.88, respectively.

### 2.6. Statistical Analysis

SPSS 26 was used for statistical analysis of the experimental data. Data from the questionnaire were first subjected to reliability analysis. A paired sample t-test was performed on the pre- and post-experimental data of participants from different departments to analyse the changes after the four weeks of intervention as reflected by their objective physiological values and subjective feelings. Next, one-way ANOVA was used to analyse any variations in each variable between the different departments before and after the intervention. Tukey was used for post-hoc analysis. Finally, an advanced analysis was made of the Pearson correlation between the changes in physiological values and subjective feelings. The significance level of testing was set to 0.05 for all statistical analyses. In addition, because of the small sample size in this study, the effect size (Cohen’s d) was measured based on observed data with effects being interpreted as small (d = 0.2), medium (d = 0.5), and large (d ≥ 0.8) [50].

## 3. Results

Table 2 shows the details of the data of the participant. The final valid samples comprised 26 nursing staff (four participants left the experiment due to resignation or departmental transfer). All the participants were females aged 21–65 years, with 42.30% of them being 21–30 years old. Nursing staff in the general and palliative care departments accounted for the majority (65.39%), followed by the ObGyn department (19.23%) and ICU (15.38%). In terms of their working experience, they ranged from being new staff (less than one year) to senior staff (more than 21 years). More than half (53.84%) of the participants had less than 15 years of experience. Most participants were university degree holders (57.69%).

### 3.1. Effect of the Intervention on Physiological Measures

The results of physiological measures on nursing staff after the diffusion of bergamot oil are shown in Table 3. There was no significant difference in all indicators, showing no obvious increment of stress between pre-post treatment.

Out of the four different departments of nursing staff, 10 participants were in the general department, seven were in the palliative care department, five were in the ObGyn department, and four were in the ICU. There were no significant changes in SYS, DIA, HR, and HRV before and after the intervention for the four departments. If we compare the nursing staff in four departments, there were increments of mean HRV from the general and ObGyn departments and a reduction from ICU and palliative care departments. The mean increased by 38.1 points in the general department between pre-intervention (M = 98.3, SD = 46.1) and post-intervention (M = 136.4, SD = 104.8), which was the highest increase for the four departments. This indicated that the participants there were more resilient to stress. For the palliative care department, the mean decreased by 27.4 points between pre-intervention (M = 79.7, SD = 48.2) and post-intervention (M = 52.3, SD = 30.5). This reduction was the greatest among the four departments, indicating additional fatigue (Figure 3).

The mean LF (%) has a significant increase of 25.75% (t = −9.69, *p* < 0.001) in ICU departments. The difference was significant, indicating an increase in the level of stress. Similarly, the mean HF (%) for the ICU participants significantly decreased by 25.75% between pre-intervention (M = 47.5, SD = 9.71) and post-intervention (M = 21.75, SD = 7.93). The mean LF/HF of participants from the ICU increased the most: 3.11 points between pre-intervention (M = 1.17, SD = 0.42) and post-intervention (M = 4.28, SD = 2.59). The difference was of borderline significance (t = −2.75, *p* < 0.05), which represented greater stress. There was a decrease in the level of relaxation, which was consistent with the analysis of LF%, HF% changes. However, there was no significant difference for the participants from the other three departments. As shown in Figure 4, the results of the one-way ANOVA analysis showed that the changes in LF% (*p* < 0.05), HF% (*p* < 0.05), and LF/HF (*p* < 0.05) were significantly different amount four departments. A large Cohen’s effect was found for LF% (d = 0.818), HF% (d = 0.818), and Ratio (d = 0.822) indicators in four groups of nursing departments. Post-hoc analysis using Tukey confirmed that the pre- and post-experimental changes in LF%, HF% from the ICU were significantly greater than the palliative care department (M = −31.03, *p* < 0.05). For LF/HF ratio, the largest changes were seen in the ICU participants. The larger the ratio, the greater the regulation by the sympathetic nervous system. Therefore, the poor effect of aromatherapy by bergamot oil was found to alleviate the symptoms of physical stress when nursing staff faces heavy workload.

### 3.2. Effect of the Intervention on Subjective Measures

The results of the paired t-test are shown in Table 4. The total score of work stress and four subscales were decreased after diffusion of bergamot, and a significant difference was found in work concerns. A small-to-moderate effect (Cohen’s d between 0.20 and 0.79) was observed in work concerns, work fatigue, and personal burnout. The mean work concerns reduced significantly by 3.92 points between pre- and post-intervention (*p* < 0.05). In addition, the work fatigue scores significantly decreased 3.23 points after the diffusion (*p* < 0.05). For the subscale, a significant decrease in personal burnout was observed. The change in personal burnout was −2.19 (*p* < 0.01).

The pre-post results for the four departments’ nursing staff showed different effects of bergamot oil on work stress release. Figure 5 shows the mean score of different subscales pre- and post-intervention for different nursing staff departments. In the general department, the mean values of all subscales were reduced and have a significant difference regarding work concerns (t = 2.615, d = 0.827, *p* < 0.05) and personal burnout (t = 2.250, d = 0.711, *p* < 0.05). The scores of work concerns decreased by 8.4 points between pre-intervention (M = 48.0, SD = 16.38) and post-intervention (M = 39.6, SD = 14.75). For personal burnout, 2.4 points were reduced between pre (M = 16.6, SD = 2.99) and post-intervention (M = 14.2, SD = 4.66). For participants from the palliative care department, only personal response, competence, and personal burnout were shown to be affected by aromatherapy. The mean of competence decreased significantly by 6 points between pre-intervention (M = 33.71, SD = 9.62) and post-intervention (M = 27.71, SD = 9.83), and the positive effect was significant (t = 2.634, d = 0.995, *p* < 0.05). In the department of ObGyn and ICU, there were no significant differences found for each subscale before and after the aromatherapy. The score decreased for work concerns, personal burnout, and client-related burnout, showing that aromatherapy has a positive effect by relieving working fatigue, although the difference was not significant.

### 3.3. Correlation Test

Next, a correlation analysis was performed of the pre- and post-experimental physiological indicators and psychological feelings. As shown in Figure 6, a significantly negative correlation exists between work concerns perception and HRV changes (Pearson correlation = −0.393, *p* = 0.047) and HF changes (Pearson correlation = −0.400, *p* = 0.043). That is, the greater the decrease in work stress perception, the greater the increase in feelings of relaxation on HRV and HF. Furthermore, there was a significant positive correlation between LF/HF ratio and personal burnout (Pearson correlation = 0.423, *p* = 0.031), which means the greater reduction in LF/HF ratio, the greater alleviation in personal burnout. Thus, the physiological and psychological responses were consistent.

## 4. Discussion

This experiment was conducted during the COVID-19 pandemic to examine the long-term efficacy of aromatherapy to regulate the work-related stress of nursing staff in different departments. The results indicated that participants from different departments had varying responses regarding the two evaluation methods of objective physiological indicators and subjective feelings. In this study, the HRV indicators had no significant changes after the aromatherapy. However, when dealing with subjective cognition, the improved stress perception of work concerns and fatigue conditions of personal burnout were shown. This is consistent with the results of Ni et al. [51], who reported an obvious improvement in psychological assessment and weak changes in physiological parameters after a bergamot oil treatment. However, Chang et al. [26] and Liu et al. [27] indicated that diffused bergamot oil has a positive effect on alleviating HRV indicators. These contradictory results may be explained by the detection time of physiological indicators, with the above two studies assessing HRV parameters immediately after the end of the intervention. A previous study indicated that the duration of aromatherapy effects on biological response is favourable for short periods [52]. In our study, we assessed the physiological results after one day of the intervention, which caused some unpredictable noise in the measurement. The diffusion time from Chang et al. [26], Liu et al. [27], and Ni et al. [51] were 60 min, 15 min, and 30 min, respectively. However, the experimental design during the essential intervention just asked the participants to sit in front of the diffuser and rest, which was different from our study. Nevertheless, the significant correlation coefficient between the psychological response and HRV indicated that the physiologically alleviated effect was present but not obviously so. The results were consistent with Lo et al.’s [49] findings, the measurement of HRV and CBI had significant associations that can be applied to assess occupational burnout. As a result, the aromatherapy of bergamot oil was effective in releasing mental stress and fatigue.

This study investigated the effect of bergamot oil on nurses working in different departments. The participants from the general department were found to have significantly improved psychological perception. They also showed positive improvements in one of their physiological indicators (HRV) and work conditions. There was no significant difference in the psychological feelings of the participants from the ICU, but there were significant negative differences in their physiological indicators (HF, HF%, and LF%). Moreover, the nursing staff from ICU had the greatest changes in LF/HF ratio, indicating increased pressure. After four weeks of intervention, the ranking of the efficacy of aromatherapy for nursing staff in the various departments in terms of duration and magnitude was as follows: general department > ObGyn department > palliative care department > ICU. Notably, the ICU nursing staff experienced increased stress.

According to Jang et al. [6], nursing staff working in the ER and ICU have higher levels of physical and mental fatigue. González-Gil et al. [12] conducted a questionnaire survey and found that ICU and ER nursing staff were prone to psychological and emotional issues in the short and medium terms because of the stress that they experienced during the COVID-19 pandemic. The conclusion and results of the clinical experiment of this study were consistent with those findings.; the stress levels within the ICU were significantly higher than that in the other departments. In general, aromatherapy had a partial effect on improving the psychological feelings of nursing staff in the general department. For nursing staff in the ObGyn and palliative care departments, the improvement effects were within the threshold. However, for the ICU nursing staff, the increment of stress exceeded the improvement effects of aromatherapy. These results show that even if physical fatigue cannot be relieved, the psychological perception can be alleviated after the diffused bergamot essential oil. This result was consistent with Liu et al.’s finding that aromatherapy treatment had a weak effect on those with a heavy workload.

This study has three main limitations. First, a control group was not included to avoid affecting the work of the nursing staff, considering that their workload had already increased rapidly during the COVID-19 pandemic. Moreover, only some of the hospital’s departments were involved. Consequently, the number of samples collected was small, and the data might be limited in some respects. Although the effect size we calculated in the results shows that the unbalance participants have acceptable power to support the hypothesis, further trials with balance sample sizes and a blinded design are suggested to obtain more accurate results. Second, because of the COVID-19 policy in hospitals, doors were not allowed to be closed so that there was adequate ventilation. Thus, the placement of the diffusers within the nursing stations of each department is considered a half-open space. The different placements might have affected the efficacy of aromatherapy. Third, the physiological measurements in our experiment were only recorded before and after the intervention. However, because biological data changes continuously over time, further exploration is needed to observe the efficacy trend of aromatherapy using the regular time-series method during the experiments.

## 5. Conclusions

We examined the effect of aromatherapy using bergamot oil to alleviate the work stress and fatigue of nursing staff during the COVID-19 pandemic. The results show that after the four-week intervention, the response of psychological stress and fatigue significantly decreased. However, there was limited improvement in physical stress when facing a greater workload. Therefore, the effect of the diffusion in different departments showed a significant improvement in general departments, an increase in changes in mental stress in ObGyn and palliative care departments, and a non-effect in ICU. This study provides practical results about aromatherapy stress reduction applied during the pandemic on the first-line medical staff.

For future research, it is suggested that the follow-up on the long-term impacts can be further explored to observe the efficacy trend of aromatherapy under severe and post-COVID-19 pandemic conditions, thus ensuring the sustainability of nursing staff mental health.

## Figures and Tables

**Figure 1 healthcare-11-00157-f001:**
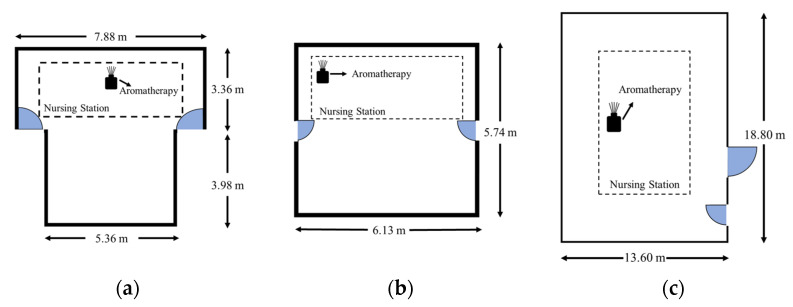
Layout of the nursing station in each department and the diffuser placement location: (**a**) General and palliative care department; (**b**) ObGyn department; and (**c**) ICU.

**Figure 2 healthcare-11-00157-f002:**
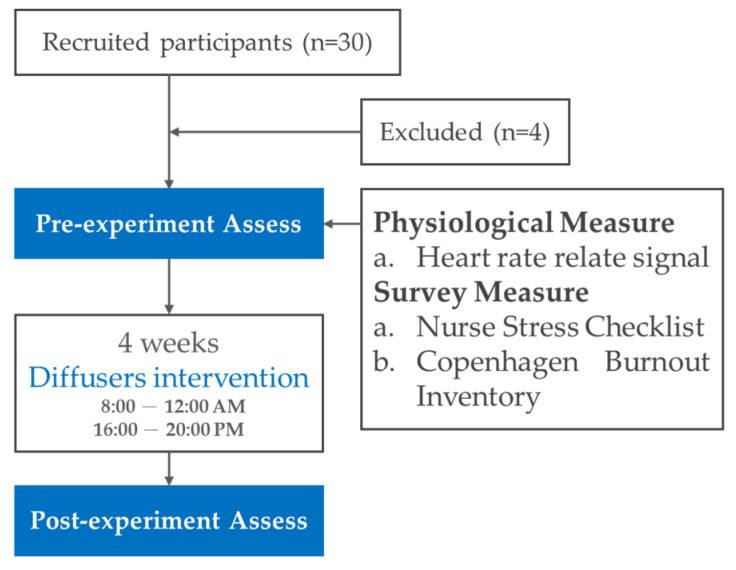
Experiment flow chart.

**Figure 3 healthcare-11-00157-f003:**
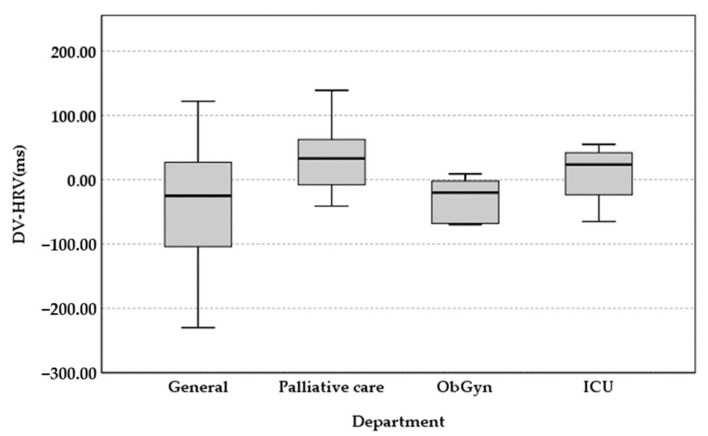
Pre-post difference value of HRV indicators.

**Figure 4 healthcare-11-00157-f004:**
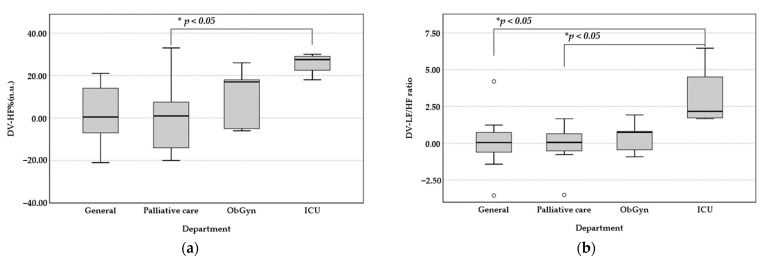
Pre-post difference value of physiological indicators: (**a**) HF%; (**b**) LF/HF ratio. (* *p* < 0.05).

**Figure 5 healthcare-11-00157-f005:**
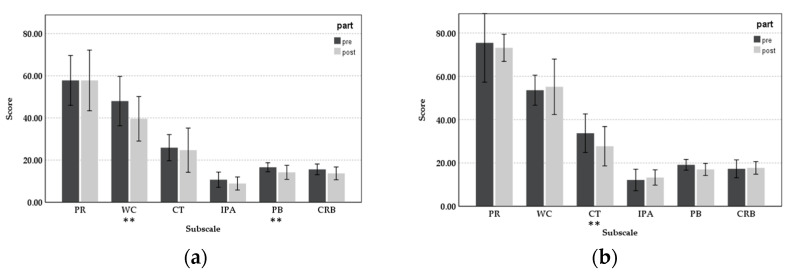
Subscale scores of NSC and CBI pre-post intervention: (**a**) General department; (**b**) Palliative care department; (**c**) ObGyn department; and (**d**) ICU (** *p* < 0.01).

**Figure 6 healthcare-11-00157-f006:**
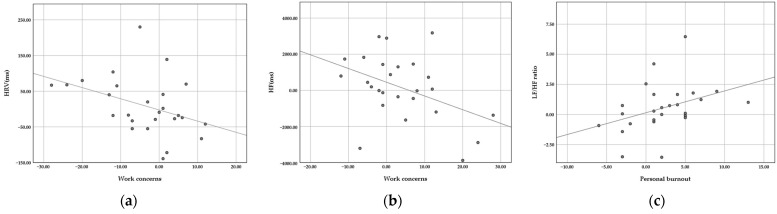
Correlation scatter plot of physiological and psychological responses: (**a**) HRV-Work concerns; (**b**) HF-Work concerns; and (**c**) LF/HF ratio-Personal burnout.

**Table 1 healthcare-11-00157-t001:** Physiological indicators and their clinical significance.

Indicator	Unit	Clinical Significance
SYS	mmHg	Systolic blood pressure
DIA	mmHg	Diastolic blood pressure
HR	BPM	Average heartbeat (beats per minute)
HRV	ms	Indicator for total activities of autonomic nervous system
HF	ms^2^	Frequency of 0.15–0.4 Hz that reflects activities of the parasympathetic nervous system
LF	ms^2^	Frequency of 0.04–0.15 Hz that reflects activities of the sympathetic nervous system
HF	%	Normalised HF, which is a quantitative indicator of activities of the parasympathetic nervous system
LF	%	Normalised LF, which is a quantitative indicator of activities of the sympathetic nervous system
LF/HF		Ratio of LF to HF, reflects the regulatory responses of the sympathetic versus parasympathetic nervous systems

Source: Task Force of the European Society of Cardiology [43].

**Table 2 healthcare-11-00157-t002:** Data of the participant information.

Demographics	Categories	Frequency	Percentage (%)
Sex	Female	26	100
Age	21–30	11	42.30
	31–40	5	19.23
	41–50	9	34.62
	>51	1	3.85
Department	General	10	38.47
	Palliative care	7	26.92
	ObGyn	5	19.23
	ICU	4	15.38
Work experience	<1 year	2	7.69
	1–15 years	12	46.15
	>15 years	12	46.15
Education level	Polytechnic diploma	8	30.77
	University degree	15	57.69
	Post-graduate degree	3	11.54
Nursing grade	N1	3	11.54
	N2	9	34.62
	N3	6	23.08
	N4	5	19.23
	Nurse Practitioner	3	11.54

**Table 3 healthcare-11-00157-t003:** Pre-post HRV indicators of paired t-test results for all participants.

Parameter	Pre (*n* = 26)	Post (*n* = 26)	t	Cohen’s d	*p*
SYS	120.27 ± 33.59	119.96 ± 27.26	0.100	2.128	0.921
DIA	80.42 ± 16.76	80.42 ± 12.93	0.000	0.000	1.000
HR	79.84 ± 14.06	81.88 ± 10.14	−0.786	0.154	0.439
HRV	90.77 ± 39.86	100.58 ± 78.31	−0.621	0.121	0.540
LF%	59.35 ± 13.83	64.96 ± 14.70	−1.422	0.279	0.167
HF%	40.65 ± 13.83	35.04 ± 14.70	1.422	0.279	0.167
LF/HF	1.84 ± 1.25	2.39 ± 1.64	−1.368	0.267	0.183

**Table 4 healthcare-11-00157-t004:** Pre-post subjective cognition of paired t-test results for all participants.

Parameter	Pre (*n* = 26)	Post (*n* = 26)	t	Cohen’s d	*p*
Total Score of NSC	145.15 ± 32.66	138.27 ± 36.30	1.651	0.299	0.069
Personal response (PR)	61.81 ± 18.90	59.50 ± 19.07	0.638	0.125	0.215
Work concerns (WC)	46.00 ± 13.93	42.08 ± 14.83	1.982	0.389	0.029 *
Competence (CT)	29.69 ± 8.74	28.88 ± 12.26	−0.459	0.091	0.325
Incompleteness personal arrangement (IPA)	10.92 ± 4.69	10.50 ± 4.32	0.441	0.086	0.332
Total Score of CBI	33.54 ± 5.45	30.31 ± 7.73	2.241	0.439	0.017 *
Personal burnout (PB)	17.65 ± 3.51	15.46 ± 4.57	2.652	0.525	0.007 **
Client-related burnout (CRB)	15.88 ± 3.56	14.85 ± 3.82	1.281	0.251	0.106

* *p* < 0.05, ** *p* < 0.01.

## Data Availability

Data sharing not applicable.

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
