# Peer review of "Efficacy of Aromatherapy at Relieving the Work-Related Stress of Nursing Staff from Various Hospital Departments during COVID-19"

_healthcare, 2023, doi:10.3390/healthcare11020157_

Round 1

Reviewer 1 Report

The biggest issue with this manuscript is that the study design is unclear. Is it a clinical trial, a quasi experimental?

The methodology section should be clear enough to present the study design the sample size, sample assignment, randomization, and ... 

Author Response

Response to Reviewer 1 Comments

1.     The biggest issue with this manuscript is that the study design is unclear. Is it a clinical trial, a quasi-experimental?

[Further explain],

Thank you for this comment. We have revised the content of “2.3. Experimental procedures” to explain the study type of this research.

[Modification]

The content of “2.3. Experimental procedures” has been revised from Ln 135-136 of page 3 to further explain as follows.

A quasi-experiment was conducted to investigate the effect of bergamot oil on nursing staff.

2.     The methodology section should be clear enough to present the study design the sample size, sample assignment, randomization, and ...

[Further explain]

We thank for Reviewer’s comment. We have elaborated the sample size on "2.1. Participants" to expand our experimental design. For the sample assignment and randomization problems, just like we mentioned in revised question 1, our study was a quasi-experiment and try to investigate the stress-released effect after aromatherapy on nursing staff. Due to the limitation of COVID-19, we only conducted the effect before and after the intervention of bergamot diffusion. Hence, we do not consider the control group in this study and explain it in "4. Discussion" on Ln 346 to 348 of page 10.

Reviewer 2 Report

1.        Please replace “paired T-test” with “paired t-test” throughout the manuscript.

2.        The quality of figures needs to watch out, especially for Fig. 3 & 4.

3.        There are many typos. The English editing service is strongly recommended to improve the readability. For example, Ln 238, in Table 4, Ln 269.

4.        As the authors mentioned, this study was conducted in an open space due to the Covid-19 situation. In Ln 124-126, “ The nature of the nursing staff’s work required them to move around the entire space. Thus, the diffusers were placed at the nursing stations to ensure that all the personnel could experience the intervention.    

-       How to verify that the participants have “experienced” the intervention? Do sight, hearing, or smell cause the experience?

5.        Ln 120. Why “five drops” of pure bergamot essential oil were dropped into the diffuser? Can any reference be used to support the setting? In my opinion, the description is too subjective and hard to control in the experiment. 

6.        Figure 1. How to determine the place for placing the diffusers? The diffusers were put in different positions in the different departments. Moreover, the sizes of the three spaces are very different. This may become a factor in influencing the results of the study. 

7.        Ln 49-53 and Ln 140-141. Please revise the sentences.

8.        One of my major concerns is using one-way ANOVA to analyze the difference between the four departments. The sample of the four departments is ten from the general department, seven from the Palliative care department, five from the ObGyn department, and four from the ICU. The number of samples is relatively less, although the authors have mentioned it as a limitation. Most of the new insights and findings of the study were related to the difference between the four departments. I am afraid the small sample sizes caused the significant differences. Please address some reasons or details. Moreover, the effect size should be provided.

9.        Ln 173. “… namely, personal fatigue, work-related fatigue, and client fatigue”. Please keep consistency throughout the manuscript to increase readability. For example, in Table 4, it became work fatigue, personal burnout, and client-related burnout. Additionally, in Ln 176-178, the authors mentioned that only two measures were used in the study due to reliability issues. It is very confusing about the presentation in Table 4 (still three measures). 

Author Response

Response to Reviewer 2 Comments

1.      Please replace “paired T-test” with “paired t-test” throughout the manuscript.

[Further explain]

We have revised the sentence in our manuscript.

[Modification]

All the text “paired T-test” has been replaced with “paired t-test” in “3. Result”.

2.     The quality of figures needs to watch out, especially for Fig. 3 & 4.

[Further explain]

We thank you for the suggestion. We have re-shape the figures to easily present the data.

3.     There are many typos. The English editing service is strongly recommended to improve the readability. For example, Ln 238, in Table 4, Ln 269.

[Further explain]

We have corrected it according to your suggestion.

[Modification]

a. The content of Ln 238 has been revised from Ln 251 on page 7

Figure 4. Pre-post difference value of physiological indicators: (a) HF%; (b) LF/HF ratio

b. The parameter name of Table 4 has been revised.

c. The content of Ln 238 has been revised.

4.     As the authors mentioned, this study was conducted in an open space due to the Covid-19 situation. In Ln 124-126, “The nature of the nursing staff’s work required them to move around the entire space. Thus, the diffusers were placed at the nursing stations to ensure that all the personnel could experience the intervention.

How to verify that the participants have “experienced” the intervention? Do sight, hearing, or smell cause the experience?

[Further explain]

We thank you for providing these insights. The experimental place in our study is a half-open area, more precisely, still in the building but all the windows and doors have opened (limitation on Ln 353 to 356 of page 10). In our case, the staffs take most of the time to finish work in the nursing station. Thus, to keep the greatest aromatherapy effect for nursing staff, the diffusers were placed at the nursing counter to ensure all staff could smell the bergamot essential oil.

We have revised the content of “2.2. Experimental design” to explain the setting of this research.

[Modification]

The content of “2.2. Experimental design” has been revised from Ln 127 to 129 of page 3 to further explain as follows.

The nature of the nursing staff’s work required them to move around the entire space; however, most of their main work was done in the nursing station.

5.     Ln 120. Why “five drops” of pure bergamot essential oil were dropped into the diffuser? Can any reference be used to support the setting? In my opinion, the description is too subjective and hard to control in the experiment. 

[Further explain]

Thank you for this comment. We have revised the content of “2.2. Experimental design” to explain the setting of this research.

[Modification]

The content of “2.2. Experimental design” has been revised from Ln 119 to 125 of page 3 to further explain as follows.

The scent was 100% pure bergamot peel essential oil produced by AndZen Co., Australia, and mainly contained limonene (< 46%), linalyl acetate (< 35%), and linalool (< 23%). An ultrasonic diffuser was used for aroma evaporation (Rose Crown Co., Australia), which only diffuses the pure essential oil molecules without water dilution. For each treatment session, five drops of pure bergamot oil were added, which make the diffuser constantly work for 4 hours and the aroma molecules were dispersed into the environment through vibration.

6.     Figure 1. How to determine the place for placing the diffusers? The diffusers were put in different positions in the different departments. Moreover, the sizes of the three spaces are very different. This may become a factor in influencing the results of the study.

[Further explain]

Thank you for providing these insights. We have elaborated the limitation on Ln 353 to 356 of page 10 to expand our consideration. For the placement of the diffusers, just like we mentioned in revised question 4, to keep the greatest aromatherapy effect for nursing staff and avoid affecting the patients, the diffusers were placed at the back side of the nursing counter.
We have revised the content of “2.2. Experimental design” and hope these revisions provide a more detail setting for this research.

[Modification]

The content of “2.2. Experimental design” has been revised from Ln 130 to 131 of page 3 to further explain as follows.

In addition, all the diffusers were placed at the back side of the nursing counter to prevent undesirable effects on patients.

7.     Ln 49-53 and Ln 140-141. Please revise the sentences.

[Further explain]

Thank you for this comment we have revised the sentences to make the expression clearer.

[Modification]

a. The content of Ln 49-53 has been revised from Ln 49 to 52of on page 2.

González-Gil et al. [12] conducted a questionnaire survey among the ICU and ER nursing staff during the peak of the COVID-19 pandemic and found that they were prone to psychological and emotional issues in the short and medium terms, because of their concerns over viral infections.

b. The content of Ln 140-141 has been revised from Ln 146 to 147of page 2.

Dzedzickis et al. [40] defined the physiological signals intended for recognition of emotions as satisfying the condition of ‘non-conscious control’.

8.     One of my major concerns is using one-way ANOVA to analyze the difference between the four departments. The sample of the four departments is ten from the general department, seven from the Palliative care department, five from the ObGyn department, and four from the ICU. The number of samples is relatively less, although the authors have mentioned it as a limitation. Most of the new insights and findings of the study were related to the difference between the four departments. I am afraid the small sample sizes caused the significant differences. Please address some reasons or details. Moreover, the effect size should be provided.

[Further explain]

We thank for Reviewer’s comment. The unbalanced participants may cause poor statistical results. We have added the "effect size" in all statistical analyses on "3.1. Effect of the intervention on Physiological Measures", "3.2. Effect of the intervention on Subjective Measures", and “Discussion” to make the results more reliable.

[Modification]

a. The content of “2.6. Statistical Analysis” has been revised from Ln 195 to 198 on page 5.

In addition, because of the small sample size in this study, the effect size (Cohen's d) was measured based on observed data with effects being interpreted as small (d = 0.2), medium (d = 0.5), and large (d ≥ 0.8) [50].

b. The content of “3.1. Effect of the intervention on Physiological Measures” has been revised from Ln 243 to 245 on page 7 and an updated Table on Table 3.

A large Cohen’s effect was found for LF%(d = 0.818), HF%(d = 0.818), and Ratio(d = 0.822) indicators in four groups of nursing departments.

c. The content of “3.2. Effect of the intervention on Subjective Measures” has been revised from Ln 256 to 257 on page 7 and an updated Table on Table4.

A small to moderate effect (Cohen's d between 0.20 and 0.79) was observed in work concerns, work fatigue, and personal burnout.

d. The content of “Discussion” has been revised from Ln 350 to 353 on page 10.

Although the effect size we calculated in the results shows the unbalance participants have acceptable power to support the hypothesis, further trials with balance sample sizes and a blinded design are suggested to obtain more accurate results.

9.     Ln 173. “…namely, personal fatigue, work-related fatigue, and client fatigue”. Please keep consistency throughout the manuscript to increase readability. For example, in Table 4, it became work fatigue, personal burnout, and client-related burnout. Additionally, in Ln 176-178, the authors mentioned that only two measures were used in the study due to reliability issues. It is very confusing about the presentation in Table 4 (still three measures).

[Further explain]

We thank for Reviewer’s comment The original indicator “Work fatigue” means the sum score of “personal burnout” and “Client-related burnout”. To avoid confusing other readers, we have revised the indicator name to make the expression clearer.

[Modification]

a. The CBI dimensions names of “2.5. Subjective Measures” in Ln 173 have been revised from Ln 178 to 180 on page 5.

It included 19 questions and covered three dimensions for evaluation, namely, personal burnout, work-related burnout, and client-related burnout.

b. The content of “2.5. Subjective Measures” in Ln 176-178 has been revised from Ln 183 to 184 of page 5.

In this study, we only investigated questions about the subscales of personal burnout and client-related burnout, and the sum score of two questions.

c. The “subjective measures title” in “3.2. Effect of the intervention on Subjective Measures” has been revised from “Work stress” and “Work fatigue” to “Total Score of NSC” and “Total Score of CBI”, respectively.

Reviewer 3 Report

An interesting study on the stress alleviating effect of a pleasant essential oil, bergamot.

I have two major concerns about the structure of the study.

1)      The low number of participants when comparing the effects in the 4 different nursing units. 26 subjects altogether is acceptable but I don’t think much consequence can be drawn from 4-5-7 participants from different units. In the limitations section though it is stated that the number of samples collected was small – this could be further stressed.

2)      I am fine with the 4-week long experiment but using aromatherapy for 4 hours (or even 2x4 hours if the same persons were still working) seems too long that may lead to a counterproductive effect. With the essential oils the less is often more, to put it this way. Why did you choose a 4-hour long interval? How long was the intervention interval in references 50, 26, 27, 51 that you compare your study to?

Others:

-          Beside the localisation of the diffusers some information should be provided on the concentration or concentration range of the bergamot oil in the air, the origin of the bergamot oil, and the type and manufacturer of the diffuser.

-          English should be improved, especially in the Results and the second part of the Discussion and the Conclusions sections.

-          2 small typos: line 187 – Pearson correlation, line 242 – bergamot.

Author Response

Response to Reviewer 3 Comments

1.     The low number of participants when comparing the effects in the 4 different nursing units. 26 subjects altogether is acceptable but I don’t think much consequence can be drawn from 4-5-7 participants from different units. In the limitations section though it is stated that the number of samples collected was small – this could be further stressed.

[Further explain]

We thank you for this comment. The unbalanced participants may cause poor statistical results. We have added the "effect size" in all statistical analyses on "3.1. Effect of the intervention on Physiological Measures", "3.2. Effect of the intervention on Subjective Measures", and “Discussion” to make the results more reliable.

[Modification]

a. The content of “2.6. Statistical Analysis” has been revised from Ln 195 to 198 on page 5.

In addition, because of the small sample size in this study, the effect size (Cohen's d) was measured based on observed data with effects being interpreted as small (d = 0.2), medium (d = 0.5), and large (d ≥ 0.8) [50].

b. The content of “3.1. Effect of the intervention on Physiological Measures” has been revised from Ln 243 to 245 on page 7 and an updated Table on Table3.

A large Cohen’s effect was found for LF%(d = 0.818), HF%(d = 0.818), and Ratio(d = 0.822) indicators in four groups of nursing departments.

c. The content of “3.2. Effect of the intervention on Subjective Measures” has been revised from Ln 256 to 257 on page 7 and an updated Table on Table4.

A small to moderate effect (Cohen's d between 0.20 and 0.79) was observed in work concerns, work fatigue, and personal burnout.

d. The content of “Discussion” has been revised from Ln 350 to 353 on page 10.

Although the effect size we calculated in the results shows the unbalance participants have acceptable power to support the hypothesis, further trials with balance sample sizes and a blinded design are suggested to obtain more accurate results.

2.     I am fine with the 4-week long experiment but using aromatherapy for 4 hours (or even 2x4 hours if the same persons were still working) seems too long that may lead to a counterproductive effect. With the essential oils the less is often more, to put it this way. Why did you choose a 4-hour long interval? How long was the intervention interval in references 50, 26, 27, 51 that you compare your study to?

[Further explain]

Thank you for providing these insights. The four hours of aromatherapy time in our study is for a simple reason, all the staff were mainly serving the patients during these two times. Moreover, we referred to research from Cho & Kwon, which indicated aromatherapy used by diffusion needs more time to reach the same effect than inhaled. From our references in the discussion, the diffusion time from Ni et al. [51], Chang et al. [26], and Liu et al. [27] were 30 min, 60 min, and 15 min, respectively. However, the experimental design during the essential intervention just asked the participants to sit in front of the diffuser and rest, which was different from our study. The discussion in this part was to prove the effect of bergamot, for the consideration of "diffusion time" has been rewritten to be more in line with your comments.

[Modification]

a. The content of “2.2. Experimental design” has been revised from line 118 to 119 of page 3 to further explain as follows.

Cho & Kwon indicated that aromatherapy by ambient delivery needs a longer length of time than inhaling to get a positive impact (for a total of 12-24 hours) [22].

b. The content of “4. Discussion” has been revised from line 312 to 315 of page 10 to further explain as follows.

The diffusion time from Chang et al. [26], Liu et al. [27], and Ni et al. [51] were 60 min, 15 min, and 30 min, respectively. However, the experimental design during the essential intervention just asked the participants to sit in front of the diffuser and rest, which was different from our study.

3.     Beside the localization of the diffusers some information should be provided on the concentration or concentration range of the bergamot oil in the air, the origin of the bergamot oil, and the type and manufacturer of the diffuser.

[Further explain]

Thank you for this comment. We have revised the content of “2.2. Experimental design” to explain the setting of this research.

[Modification]

The content of “2.2. Experimental design” has been revised from Ln 119 to 125 of page 3 to further explain as follows.

The scent was 100% pure bergamot peel essential oil produced by AndZen Co., Australia, and mainly contained limonene (< 46%), linalyl acetate (< 35%), and linalool (< 23%). An ultrasonic diffuser was used for aroma evaporation (Rose Crown Co., Australia), which only diffuses the pure essential oil molecules without water dilution. For each treatment session, five drops of pure bergamot oil were added, which make the diffuser constantly work for 4 hours and the aroma molecules were dispersed into the environment through vibration.

4.      English should be improved, especially in the Results and the second part of the Discussion and the Conclusions sections.

[Further explain]

We thank for the Reviewer’s comment and have corrected it according to your suggestion. In addition, we have asked a native English editor to revise the manuscript.

5.     2 small typos: line 187 – Pearson correlation, line 242 – bergamot.

[Further explain]

Thanks for the correction. We have replaced the wrong text in our manuscript.

[Modification]

a. The text of wrong spelling of “Person correlation” has been corrected to “Pearson correlation” in Ln 194 page 5.

b. The text of wrong spelling of “burgamot” has been corrected to “bergamot” in Ln 255 page 7.

Round 2

Reviewer 1 Report

The authors addressed most of issues properly.

Reviewer 2 Report

Thanks for the authors carefully and completely responses all my concerns. There is no more question raised from me.

Reviewer 3 Report

I see you have worked a lot on improving the manuscript, and provided clear and thorough responses to my comments and made such changes to the text and to the presentation of the study.

I accept it in the present form.